# Handling Missing Data in Decision Trees: A Probabilistic Approach

**Pasha Khosravi** [1]   **Antonio Vergari** [1]   **YooJung Choi** [1]   **Yitao Liang** [1]   **Guy Van den Broeck** [1]

## Abstract

Decision trees are a popular family of models due to their attractive properties such as interpretability and ability to handle heterogeneous data. Concurrently, missing data is a prevalent occurrence that hinders performance of machine learning models. As such, handling missing data in decision trees is a well studied problem. In this paper, we tackle this problem by taking a probabilistic approach. At deployment time, we use tractable density estimators to compute the "expected prediction" of our models. At learning time, we fine-tune parameters of already learned trees by minimizing their "expected prediction loss" w.r.t. our density estimators. We provide brief experiments showcasing effectiveness of our methods compared to few baselines.

## 1. Introduction

Decision trees for classification and regression tasks have a long history in ML and AI (Quinlan, 1986; Breiman et al., 1984). Despite the remarkable successes of deep learning and the enormous attention it attracts, trees and forests are still the preferred off-the-shelf model when in need of robust and interpretable learning on scarce data that is possibly heterogeneous (mixed continuous-discrete) in nature and featuring missing values (Chen & Guestrin, 2016; Devos et al., 2019; Prokhorenkova et al., 2018).

In this work we specifically focus on the last property, noting that while trees are widely regarded as flawlessly handling missing values, *there is no unique way to properly deal with missingness in trees* when it comes to tree induction from data (learning time) or reasoning about partial configurations of the world (deployment time).

Numerous strategies and approaches have been explored in the literature in this regard (Saar-Tsechansky & Provost,

2007; Gavankar & Sawarkar, 2015; Twala et al., 2008). However, most of these are heuristics in nature (Twala et al., 2008), tailored towards some specific tree induction algorithm (Chen & Guestrin, 2016; Prokhorenkova et al., 2018), or make strong distributional assumptions about the data, such as the feature distribution factorizing completely (e.g., mean, median imputation (Rubin, 1976)) or according to the tree structure (Quinlan, 1993). As many works have compared the most prominent ones in empirical studies (Batista & Monard, 2003; Saar-Tsechansky & Provost, 2007), there is no clear winner and ultimately, the adoption of a particular strategy in practice boils down to its availability in the ML libraries employed.

In this work, we tackle handling missing data in trees at both learning and deployment time from a principled probabilistic perspective. We propose to decouple tree induction from learning the joint feature distribution, and we leverage tractable density estimators to flexibly and accurately model it. Then we exploit tractable marginal inference to efficiently compute the *expected predictions* (Khosravi et al., 2019a) of tree models. In essence, the expected prediction of a tree given a sample with missing values can be thought of as implicitly imputing all possible completions at once, reweighting each complete sample by its probability. In such a way, we can improve the performances of already learned trees, e.g., by XGBoost (Chen & Guestrin, 2016), by making their predictions robust under missing data at deployment time.

Moreover, we show how expected predictions can also be leveraged to deal with *missing data at learning time* by efficiently training trees over the expected version of commonly-used training losses (e.g., MSE). As our preliminary experiments suggest, this probabilistic perspective delivers better performances than common imputation schemes or default ways to deal with missing data in popular decision tree implementations. Lastly, this opens several interesting research venues such as to devise probabilistically principled tree structure induction and robustness against different kinds of missingness mechanisms.

## 2. Background

We use uppercase letters ($X$) for random variables (RVs) and lowercase letters ($x$) for their assignments. Analogously,

---

[1]Department of Computer Science, University of California Los Angeles. Correspondence to: Pasha Khosravi <pashak@cs.ucla.edu>.

*Presented at the first Workshop on the Art of Learning with Missing Values (Artemiss) hosted by the $37^{th}$ International Conference on Machine Learning (ICML).* Copyright 2020 by the author(s).

we denote sets of RVs in bold uppercase ($\mathbf{X}$) and their assignments in bold lowercase ($\mathbf{x}$). We denote the set of all possible assignments to $\mathbf{X}$ as $\mathcal{X}$. We denote a *partial* assignment to RVs $\mathbf{X}^o \subset \mathbf{X}$ as $\mathbf{x}^o$ and a possible completion to it as $\mathbf{x}^m$, that is, an assignment to RVs $\mathbf{X}^m = \mathbf{X} \setminus \mathbf{X}^o$.

**Decision trees**    Given a set of input RVs $\mathbf{X}$ (features) and an RV $Y$ (target) having values in $\mathcal{Y}$, a *decision tree* $f_\Theta$ is a parameterized mapping $f_\Theta : \mathcal{X} \to \mathcal{Y}$ characterized by a pair $(\mathcal{T}, \Theta)$ where $\mathcal{T}$ is a rooted tree structure and $\Theta = \{\theta_\ell\}_{\ell \in \text{leaves}(\mathcal{T})}$ is a set of parameters equipped to the *leaves* of $\mathcal{T}$. Every non-leaf node $n$ in $\mathcal{T}$, also called a *decision node*, is labeled by an RV $X_n \in \mathbf{X}$. For a decision node $n$ the set of $k$ outgoing edges $\{(n, j)\}_{j=1}^k$ partitions $\mathcal{X}_n$, the set of values of RV $X_n$, into a set of $k$ disjoint sets $\mathcal{X}_n^1 \cup \ldots \cup \mathcal{X}_n^k = \mathcal{X}_n$, and defines a set of corresponding decision tests $\{[\![x_n \in \mathcal{X}_n^j]\!]\}_{j=1}^k$. A decision *path* path$(\ell)$ is a collection of adjacent edges from the root of $\mathcal{T}$ to leaf $\ell$. Given the above, the mapping encoded in a decision tree $(\mathcal{T}, \Theta)$ can be written as:

$$f_\Theta(\mathbf{x}) = \sum_{\ell \in \text{leaves}(\mathcal{T})} \theta_\ell \, \mathcal{I}_\ell(\mathbf{x}) \tag{1}$$

where $\mathcal{I}_\ell(\mathbf{x})$ is an indicator function that is equal to 1 if $\mathbf{x}$ "reaches" leaf $\ell$ and 0 otherwise; formally, $\mathcal{I}_\ell(\mathbf{x}) = \prod_{(n,j) \in \text{path}(\ell)} [\![x_n \in \mathcal{X}_n^j]\!]$, where $x_n$ is the assignment for RV $X_n$ in $\mathbf{x}$. The parameters attached to the leaves in $\mathcal{T}$ here represent a *hard* prediction, i.e., $\theta_\ell = y_\ell$ for some value $y_\ell \in \mathcal{Y}$ associated to leaf $\ell$. Our derivations will also hold when $f_\Theta$ encodes a *soft* predictor, e.g. for $C$-class classification, $f_\Theta : \mathcal{X} \to [0, 1]^C$. In that case, we consider a parameter vector $\boldsymbol{\theta}_\ell$ for each leaf $\ell$ comprising $C$ conditional probabilities $\theta_\ell^i = p(Y = i \mid \mathbf{x})$ for $i = 1, \ldots, C$.

In the following, we will assume RVs $\mathbf{X}$ to be discrete. This is to simplify notation and does not hinder generality: our derivations can be easily extended to mixed discrete and continuous RVs by replacing summations to integrations when needed. Note that in the discrete case, a decision node $n$ labeled by RV $X_n$ having $k$ different states, i.e., $\mathcal{X}_n = \{1, \ldots k\}$, will define $k$ decision tests for one assignment $x_n$ will be $[\![x_n = j]\!]$ for $j = 1, \ldots, k$.

**Decision forests**    Single tree models are aggregated in ensembles called *forests* (Breiman, 1996). One of the most common way to build a forest of $R$ trees is to put them in a weighted additive ensemble of the form

$$F_\Theta(\mathbf{x}) = \sum_{r=1}^R \omega_r f_{\Theta_r}(\mathbf{x}). \tag{2}$$

This is the case for ensembling techniques like bagging (Breiman, 1996), random forests (Breiman, 2001) and gradient boosting (Friedman, 2001).

**Decision trees for missing data**    Several ways have been explored to deal with missing values for decision trees both

at training and inference (test) time (Saar-Tsechansky & Provost, 2007). One of the most common approaches goes under the name of predictive value imputation (PVI) and resorts to replacing missing values before performing inference or tree induction. Among the simplest treatments to missing values in PVI, mean, median and mode imputations are practical and cheap common techniques (Rubin, 1976; Breiman, 2001); however, they make strong distributional assumptions like total independence of the feature RVs. More sophisticated (and expensive) PVI techniques cast imputation as prediction from observed features. Among these are multiple imputation with chained equations (MICE) (Buuren & Groothuis-Oudshoorn, 2010) and the use of *surrogate splits* as popularized by CART (Breiman et al., 1984).

Somehow analogous to PVI methods, the missing value treatment done by XGBoost (Chen & Guestrin, 2016) learns to predict which branch to take for a missing feature at inference time by improving some gain criterion on observed data for that feature. While this approach has been proven successful in many real-world scenarios with missing data, it *requires data to be missing at learning time* and it may *overfit* to the missingness pattern observed.

Unlike above imputation schemes, the approach introduced in C4.5 (Quinlan, 1993) replaces imputation with reweighting the prediction associated to one instance by the product of the probabilities of the missing RVs in it. While C4.5 is more distribution aware, these probabilities acting as weights are only empirical estimates from the training data, and reweighting is limited only to the missing attributes appearing in a path. This assumes that the true distribution over $\mathbf{X}$ factorizes exactly as the tree structure, which is hardly the case since the tree structure is induced to minimize some predictive loss over $Y$.

Several empirical studies showed evidence that there is no clear winner among the aforementioned approaches under different distributional and missingness assumptions (Batista & Monard, 2003; Saar-Tsechansky & Provost, 2007). In practice, the adoption of a particular strategy is dependent on the specific tree learning or inference algorithm selected, and on the availability of its implemented routines. We introduce in the next section a principled probabilistic and tree-agnostic way of treating missing values at deployment time and extend it to deal with missingness at learning time in Section 4.

## 3. Expected Predictions of Decision Trees

From a probabilistic perspective, we would like a missing value treatment to be *aware of the full distribution* over RVs $\mathbf{X}$ without committing to restrictive distributional assumptions. If we have access to the joint distribution $p(\mathbf{X})$, then clearly the best way to deal with missing values at inference

time would be *to impute all possible completions at once*, weighting them by their probabilities according to $p(\mathbf{X})$, thereby generalizing both C4.5 and PVI treatments. This is what the *expected prediction* estimator delivers.

**Definition 1** (Expected prediction). *Given a predictive model $f_\Theta : \mathcal{X} \to \mathcal{Y}$, a distribution $p(\mathbf{X})$ over features $\mathbf{X}$ and a partial assignment $\mathbf{x}^o$ for RVs $\mathbf{X}^o \subset \mathbf{X}$, the expected prediction of $f$ w.r.t. $p$ is:*

$$\mathbb{E}_{\mathbf{x}^m \sim p(\mathbf{X}^m|\mathbf{x}^o)}\left[f_\Theta(\mathbf{x}^o, \mathbf{x}^m)\right] \qquad (3)$$

*where $\mathbf{X}^m = \mathbf{X} \setminus \mathbf{X}^o$ and $f_\Theta(\mathbf{x}^o, \mathbf{x}^m) = f_\Theta(\mathbf{x})$.*

Computing expected predictions is theoretically appealing also because the delivered estimator is consistent under both MCAR and MAR missingness mechanisms, if $f$ has been trained on complete data and is Bayes optimal (Josse et al., 2019). As one would expect, however, computing Equation 3 exactly for arbitrary pairs of $f$ and $p$ is NP-hard (Khosravi et al., 2019b). Recently, (Khosravi et al., 2019a) identified a class of expressive density estimators $p$ and accurate predictive models $f$ that allows for polytime computation of the expected predictions of the latter w.r.t. the former. Specifically, probabilistic circuits (PCs) (Choi et al., 2020) with certain structural restrictions can be used as tractable density estimators to compute the expected predictions exactly for regression and to approximate them for classification, from simple models such as linear and logistic regression to their generalization as circuits (Liang & Van den Broeck, 2019). Here we extend those results to compute expected predictions for both classification and regression trees *exactly* and *efficiently* under milder distributional assumptions for $p$.

**Proposition 1** (Expected predictions for decision trees). *Given a decision tree $(\mathcal{T}, \Theta)$ encoding $f_\Theta(\mathbf{x})$, a distribution $p(\mathbf{X})$, and a partial assignment $\mathbf{x}^o$, the expected prediction of $f$ w.r.t. $p$ can be computed as follows:*

$$\mathbb{E}_{\mathbf{x}^m \sim p(\mathbf{X}^m|\mathbf{x}^o)}\left[f_\Theta(\mathbf{x}^o, \mathbf{x}^m)\right] = \frac{1}{p(\mathbf{x}^o)} \sum_{\ell \in \mathsf{leaves}(\mathcal{T})} \theta_\ell \cdot p_\ell(\mathbf{x}^o)$$

$$(4)$$

*where $p_\ell(\mathbf{x}^o) = p(\mathbf{x}^{\mathsf{path}(\ell)}, \mathbf{x}^o)$ and $\mathbf{x}^{\mathsf{path}(\ell)}$ is the assignment to the RVs in $\mathsf{path}(\ell)$ that evaluates $\mathcal{I}_\ell(\mathbf{x}') = \prod_{(n,j)\in\mathsf{path}(\ell)}[\![x'_n = j]\!]$ to 1.*

We refer to Appendix A for detailed derivations. Note that we can readily extend Equation 4 to forests of trees (cf. Equation 2) by linearity of expectations.

As the proposition suggests, we can tractably compute the exact expected predictions of a decision tree if the number of its leaves is polynomial in the input size and we can compute $p_\ell(\mathbf{x}^o)$ in polytime for each leaf $\ell$. The first condition generally holds in practice, as trees have low-depth to avoid overfitting, especially in forests, while the second one can be

easily satisfied by employing a probabilistic model guaranteeing tractable marginalization, as we need to marginalize over the RVs not in $\mathsf{path}(\ell)$. Among suitable candidates are Gaussian distributions and their mixtures for continuous data, and smooth and decomposable PCs (Choi et al., 2020) which are deep versions of classical mixture models. We employ PCs in our experiments as they are potentially more expressive than shallow mixtures and can seamlessly model mixed discrete-continuous distributions.

## 4. Expected Parameter Learning of Trees

Expected predictions provide a principled way to deal with missing values at inference time. In the following, we extend them to learn the parameters $\Theta$ of a predictive model from incomplete data as to minimize a the expectation of a certain loss w.r.t. a generative model at hand. We call this learning scenario, *expected loss minimization*.

**Definition 2** (Expected loss minimization). *Given a dataset $\mathsf{D}_{\mathsf{train}}$ over $\mathcal{X} \times \mathcal{Y}$ containing missing values for RVs $\mathbf{X}$, a density estimator $p_\Phi(\mathbf{X})$ trained on $\mathsf{D}_{\mathsf{train}}$ by maximum likelihood, and a per-sample loss function $l$, we want to find the set of parameters $\Theta$ of the predictive model $f_\Theta : \mathcal{X} \to \mathcal{Y}$ that minimizes the expected loss $\mathcal{L}(\Theta)$ defined as follows:*

$$\mathcal{L}(\Theta; \mathsf{D}_{\mathsf{train}}) = \frac{1}{|\mathsf{D}_{\mathsf{train}}|} \sum_{\mathbf{x}^o, y \in \mathsf{D}_{\mathsf{train}}} \mathbb{E}_{p_\Phi(\mathbf{X}^m|\mathbf{x}^o)}\left[l(y, f_\Theta(\mathbf{x}))\right]$$

Again, we harness the ability of the density estimator $p_\Phi$ to accurately model the distribution over RVs $\mathbf{X}$ and to minimize the loss over $f_\Theta$ as if it were trained on all possible completions for a partial configuration $\mathbf{x}^o$.

For commonly used per-sample losses, the optimal set of parameters for single decision trees can be efficiently and independently computed in closed form. This is for instance the case for the $L_2$ loss, also known as mean squared error (MSE), defined as $l_{\mathsf{MSE}}(y, f_\Theta(\mathbf{x})) := (y - f_\Theta(\mathbf{x}))^2$, which we will use in our experiments.

**Proposition 2** (Expected parameters of MSE loss). *Given a decision tree structure $\mathcal{T}$ and a training set $\mathsf{D}_{\mathsf{train}}$, the set of parameters $\Theta = \{\theta_\ell\}_{\ell \in \mathsf{leaves}(\mathcal{T})}$ that minimizes $\mathcal{L}_{\mathsf{MSE}}$, the expected prediction loss for MSE, can be found by*

$$\theta_\ell^* = \frac{\sum_{\mathbf{x}^o, y \in \mathsf{D}_{\mathsf{train}}} y \cdot p_\ell(\mathbf{x}^o)/p(\mathbf{x}^o)}{\sum_{\mathbf{x}^o, y \in \mathsf{D}_{\mathsf{train}}} p_\ell(\mathbf{x}^o)/p(\mathbf{x}^o)}$$

*for each leaf $\ell \in \mathsf{leaves}(\mathcal{T})$.*

The above equation for optimal leaf parameters can be extended to forests of trees where each tree is learned independently, e.g., via bagging. For other scenarios involving forests such as boosting refer to Appendix B.

Furthermore, a regularization term may be added to the expected loss to counter overfitting in a regression scenario,

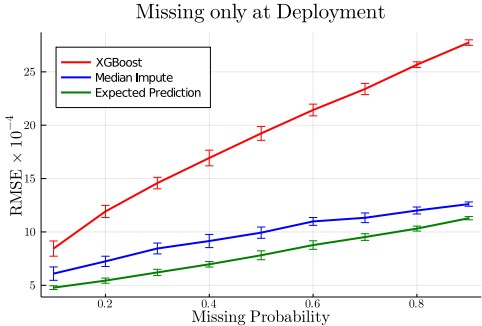 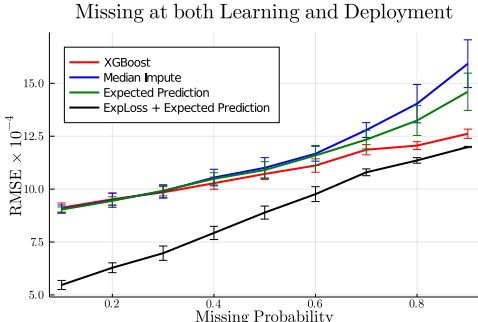

*Figure 1.* Average test RMSE (y-axis, the lower, the better) on the Insurance data for different percentages of missing values (x-axis) when missingness is only at deployment time for a forest of 5 trees (left) or both at learning and deployment time for a single tree learned with XGBoost (right). For each experiment setting, we repeat 10 times and report the average error and their standard deviation.

e.g., by penalizing the leaf parameter magnitude; this still yields close-form solutions (see Appendix A). Next, we will show the effect of tree parameter learning via expected losses on some tree structures that have been induced by popular algorithms such as XGBoost (Chen & Guestrin, 2016); that is, we will *fine tune* their parameters to optimality given their structures and a tractable density estimator for $p(\mathbf{X})$. Investigating how to blend expected loss learning in classical top-down tree induction schemes is an interesting venue we are currently exploring.

## 5. Experiments

In this section, we provide preliminary experiments to answer the following questions: (**Q1**) Do expected predictions at deployment time improve predictions over common techniques to deal with missingness for trees? (**Q2**) Does expected loss minimization improve predictions when missing values are present also at learning time?

**Setup**    We employ the Insurance dataset, in which we want to predict the yearly medical insurance costs of a patient based their personal data.[1] We consider two scenarios, when data is missing only at deployment time or also at learning time. In both cases, we assume data to be MCAR: given complete data, we make each feature be missing with probability $\pi \in \{0.1, 0.2, \ldots, 0.9\}$ each for 10 independent trials. For each setting, we learn a probabilistic circuit $p_\Phi$ on the training data as well as a decision tree or forest using the ubiquitous XGBoost.

**Methods**    For XGBoost we employ the default parameters. As a simple baseline we use median imputation, estimating the per-feature imputations on the observed portion of the training set. We employ expected predictions over the trees learned by XGBoost for dealing with missing data at deployment time. Lastly, we use expected loss to fine-tune

the XGBoost trees and use them for expected predictions at deployment time, which we denote as "ExpLoss + Expected Prediction". We measure performance by the average test root mean squared error (RMSE).

**Missing only at deployment time**    Figure 1(left) summarizes our results for **Q1**. Expected prediction outperforms XGBoost and median imputation. Notably, the reason XGBoost performs poorly is that it has not seen any missing values at learning time, in which case the "default" branch it uses in case of missing values always points to the first child. Additionally, median imputation makes the strong assumption that all the features are fully independent, which would explain why expected prediction using PCs does better.

**Missing during both learning and deployment**    Figure 1 summarizes our results for **Q2**. In this scenario, expected predictions perform on par, up to $\pi = 0.4$, with the way XGBoost treats missing values at deployment time generated from the same missingness mechanism it has been trained on. However, both methods are significantly outperformed by fine-tuning the tree parameters by the expected loss minimization. We leave for future work to investigate what happens with missingness mechanisms that differ at learning and deployment time, or when we adopt other ensembling techniques such as bagging and random forests.

## 6. Conclusion

In this work, we introduced expected predictions and expected loss minimization for decision trees and forests as a principled probabilistic way to handle missing data both at training and deployment time, while being agnostic to the tree structure or the way it has been learned. We are currently investigating how to exploit this methodology to extend tree induction schemes under different missing value mechanisms and derive consistency guarantees for the learned estimators.

---

[1] Refer to Appendix C for more information about the dataset.

## Acknowledgments

This work is partially supported by NSF grants #IIS-1943641, #IIS-1633857, #CCF-1837129, DARPA XAI grant #N66001-17-2-4032, UCLA Samueli Fellowship, and gifts from Intel and Facebook Research. The authors would like to thank Steven Holtzen for initial discussions about expected prediction for decision trees.

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

## A. Proofs

**Proposition 1** (Expected predictions for decision trees). *Given a decision tree $(\mathcal{T}, \Theta)$ encoding $f_\Theta(\mathbf{x})$, a distribution $p(\mathbf{X})$, and a partial assignment $\mathbf{x}^o$, the expected prediction of $f$ w.r.t. $p$ can be computed as follows:*

$$\mathbb{E}_{\mathbf{x}^m \sim p(\mathbf{X}^m | \mathbf{x}^o)}[f_\Theta(\mathbf{x}^o, \mathbf{x}^m)] = \frac{1}{p(\mathbf{x}^o)} \sum_{\ell \in \mathsf{leaves}(\mathcal{T})} \theta_\ell \cdot p_\ell(\mathbf{x}^o)$$

*where $p_\ell(\mathbf{x}^o) = p(\mathbf{x}^{\mathsf{path}(\ell)}, \mathbf{x}^o)$ and $\mathbf{x}^{\mathsf{path}(\ell)}$ is the assignment to the RVs in $\mathsf{path}(\ell)$ that evaluates $\mathcal{I}_\ell(\mathbf{x}') = \prod_{(n,j) \in \mathsf{path}(\ell)} [\![x'_n = j]\!]$ to 1.*

*Proof.* Let $\mathbf{X}^{\mathsf{path}(\ell)}$ be the set of RVs appearing in the decision nodes in path $\mathsf{path}(\ell)$. Then the following holds:

$$\mathbb{E}_{\mathbf{x}^m \sim p(\mathbf{X}^m | \mathbf{x}^o)}[f_\Theta(\mathbf{x}^o, \mathbf{x}^m)]$$
$$= \frac{1}{p(\mathbf{x}^o)} \mathbb{E}_{\mathbf{x}^m \sim p(\mathbf{x}^m, \mathbf{x}^o)}[f_\Theta(\mathbf{x}^o, \mathbf{x}^m)]$$
$$= \frac{1}{p(\mathbf{x}^o)} \mathbb{E}_{\mathbf{x}^m \sim p(\mathbf{x}^m, \mathbf{x}^o)} \left[ \sum_{\ell \in \mathsf{leaves}(\mathcal{T})} \theta_\ell \mathcal{I}_\ell(\mathbf{x}) \right]$$
$$= \frac{1}{p(\mathbf{x}^o)} \sum_{\ell \in \mathsf{leaves}(\mathcal{T})} \theta_\ell \, \mathbb{E}_{\mathbf{x}^m \sim p(\mathbf{x}^m, \mathbf{x}^o)}[\mathcal{I}_\ell(\mathbf{x})]$$
$$= \frac{1}{p(\mathbf{x}^o)} \sum_{\ell \in \mathsf{leaves}(\mathcal{T})} \theta_\ell \, \mathbb{E}_{\mathbf{x}^m \sim p(\mathbf{x}^m, \mathbf{x}^o)} \left[ \prod_{(n,j) \in \mathsf{path}(\ell)} [\![x_n = j]\!] \right]$$
$$= \frac{1}{p(\mathbf{x}^o)} \sum_{\ell \in \mathsf{leaves}(\mathcal{T})} \theta_\ell \, p(\mathbf{x}^{\mathsf{path}(\ell)}, \mathbf{x}^o)$$

$\square$

Before proving Proposition 2, let us first introduce a useful lemma.

**Lemma 1** (Expected squared predictions). *Given a decision tree structure $(\mathcal{T}, \Theta)$ encoding $f_\Theta(\mathbf{x})$, a distribution $p(\mathbf{X})$ and a partial assignment $\mathbf{x}^o$ the expected squared prediction of $f$ w.r.t. $p$ can be computed as follows:*

$$\mathbb{E}_{p_\phi(\mathbf{X}^m | \mathbf{x}^o)}[f_\Theta^2(\mathbf{x})] = \frac{1}{p(\mathbf{x}^o)} \sum_{\ell \in \mathsf{leaves}(\mathcal{T})} \theta_\ell^2 p_\ell(\mathbf{x}^o).$$

*Proof.*

$$\mathbb{E}_{p_\phi(\mathbf{X}^m | \mathbf{x}^o)}[f_\Theta^2(\mathbf{x})] = \frac{1}{p(\mathbf{x}^o)} \mathbb{E}_{p_\phi(\mathbf{X}^m, \mathbf{x}^o)}[f_\Theta^2(\mathbf{x})]$$
$$= \frac{1}{p(\mathbf{x}^o)} \mathbb{E}_{p_\phi(\mathbf{X}^m, \mathbf{x}^o)} \left[ \left( \sum_{\ell \in \mathsf{leaves}(\mathcal{T})} \theta_\ell \mathcal{I}_\ell(\mathbf{x}) \right)^2 \right]$$
$$= \frac{1}{p(\mathbf{x}^o)} \mathbb{E}_{p_\phi(\mathbf{X}^m, \mathbf{x}^o)} \left[ \sum_{\ell \in \mathsf{leaves}(\mathcal{T})} \theta_\ell^2 \mathcal{I}_\ell(\mathbf{x}) \right] \quad (5)$$

$$= \frac{1}{p(\mathbf{x}^o)} \sum_{\ell \in \mathsf{leaves}(\mathcal{T})} \theta_\ell^2 \cdot \mathbb{E}_{p_\phi(\mathbf{X}^m, \mathbf{x}^o)}[\mathcal{I}_\ell(\mathbf{x})]$$
$$= \frac{1}{p(\mathbf{x}^o)} \sum_{\ell \in \mathsf{leaves}(\mathcal{T})} \theta_\ell^2 \cdot p_\ell(\mathbf{x}^o). \quad (6)$$

where Eq. 5 follows from the fact that $\mathcal{I}_\ell(\mathbf{x}) \cdot \mathcal{I}_{\ell'}(\mathbf{x}) = 0$ iff $\ell \neq \ell'$ and from the idempotence of indicator functions ($\mathcal{I}_j^2(\mathbf{x}) = \mathcal{I}_j(\mathbf{x})$), whereas Eq. 6 follows the proof of Proposition 1. $\square$

**Proposition 2** (Expected parameters of MSE loss). *Given a decision tree structure $\mathcal{T}$ and a training set $\mathsf{D}_{\mathsf{train}}$, the set of parameters $\Theta = \{\theta_\ell\}_{\ell \in \mathsf{leaves}(\mathcal{T})}$ that minimizes $\mathcal{L}_{\mathsf{MSE}}$, the expected prediction loss for MSE, can be found by*

$$\theta_\ell^* = \frac{\sum_{\mathbf{x}^o, y \in \mathsf{D}_{\mathsf{train}}} y \cdot p_\ell(\mathbf{x}^o)/p(\mathbf{x}^o)}{\sum_{\mathbf{x}^o, y \in \mathsf{D}_{\mathsf{train}}} p_\ell(\mathbf{x}^o)/p(\mathbf{x}^o)}$$

*for each leaf $\ell \in \mathsf{leaves}(\mathcal{T})$.*

*Proof.* First, the expected MSE loss can be expressed as the following:

$$\mathcal{L}_{\mathsf{MSE}}(\Theta; \mathsf{D}_{\mathsf{train}})$$
$$= \frac{1}{|\mathsf{D}_{\mathsf{train}}|} \sum_{\mathbf{x}^o, y \in \mathsf{D}_{\mathsf{train}}} \mathbb{E}_{p_\phi(\mathbf{X}^m | \mathbf{x}^o)} \left[ (y - f_\Theta(\mathbf{x}))^2 \right]$$
$$= \frac{1}{|\mathsf{D}_{\mathsf{train}}|} \sum_{\mathbf{x}^o, y \in \mathsf{D}_{\mathsf{train}}} \left( y^2 - 2y \mathbb{E}_{p_\phi(\mathbf{X}^m | \mathbf{x}^o)}[f_\Theta(\mathbf{x})] \right.$$
$$\left. + \mathbb{E}_{p_\phi(\mathbf{X}^m | \mathbf{x}^o)}[f_\Theta^2(\mathbf{x})] \right).$$

To optimize this loss, we consider its partial derivative w.r.t. a leaf parameter $\theta_\ell$. Using Equation 3 and the fact that gradient is a linear operator, we have:

$$\frac{\partial \mathbb{E}_{p_\phi(\mathbf{X}^m | \mathbf{x}^o)}[f_\Theta(\mathbf{x})]}{\partial \theta_\ell} = \frac{p_\ell(\mathbf{x}^o)}{p(\mathbf{x}^o)}.$$

Similarly, the partial derivative of expected squared prediction in Lemma 1 is:

$$\frac{\partial \mathbb{E}_{p_\phi(\mathbf{X}^m | \mathbf{x}^o)}[f_\Theta^2(\mathbf{x})]}{\partial \theta_\ell} = \frac{2\theta_\ell p_\ell(\mathbf{x}^o)}{p(\mathbf{x}^o)}.$$

Therefore, the partial derivative of expected MSE loss w.r.t. a leaf parameter $\theta_\ell$ can be computed as follows:

$$\frac{\partial \mathcal{L}_{\mathsf{MSE}}}{\partial \theta_\ell}$$
$$= \frac{1}{|\mathsf{D}_{\mathsf{train}}|} \sum_{\mathbf{x}^o, y \in \mathsf{D}_{\mathsf{train}}} \left( -2y \frac{\partial \mathbb{E}_{p_\phi(\mathbf{X}^m | \mathbf{x}^o)}[f_\Theta(\mathbf{x})]}{\partial \theta_\ell} \right.$$

$$+ \frac{\partial \mathbb{E}_{p_\phi(\mathbf{X}^m | \mathbf{x}^o)}\left[f_\Theta^2(\mathbf{x})\right]}{\partial \theta_\ell}\Bigg)$$

$$= \frac{1}{|\mathsf{D}_{\mathsf{train}}|} \sum_{\mathbf{x}^o, y \in \mathsf{D}_{\mathsf{train}}} \left(-2y \frac{p_\ell(\mathbf{x}^o)}{p(\mathbf{x}^o)} + \frac{2\theta_\ell p_\ell(\mathbf{x}^o)}{p(\mathbf{x}^o)}\right).$$

Then its gradient w.r.t. the parameter vector $\boldsymbol{\theta} = [\theta_{\ell_1}, \ldots, \theta_{\ell_L}]$, with $L = |\mathsf{leaves}(\mathcal{T})|$, can be written in matrix notation as:

$$\nabla_{\boldsymbol{\theta}}\mathcal{L} = \frac{2}{|\mathsf{D}_{\mathsf{train}}|} \sum_{\mathbf{x}^o, y \in \mathsf{D}_{\mathsf{train}}} \begin{bmatrix} (\theta_{\ell_1} - y)\, p_{\ell_1}(\mathbf{x}^o)/p(\mathbf{x}^o) \\ \vdots \\ (\theta_{\ell_L} - y)\, p_{\ell_L}(\mathbf{x}^o)/p(\mathbf{x}^o) \end{bmatrix}$$

Hence, by setting $\nabla_{\boldsymbol{\theta}}\mathcal{L} = \mathbf{0}$ we can easily retrieve that the optimal parameter vector is:

$$\boldsymbol{\theta}^* = \begin{bmatrix} \frac{\sum_{\mathbf{x}^o, y \in \mathsf{D}_{\mathsf{train}}} y \cdot p_{\ell_1}(\mathbf{x}^o)/p(\mathbf{x}^o)}{\sum_{\mathbf{x}^o, y \in \mathsf{D}_{\mathsf{train}}} p_{\ell_1}(\mathbf{x}^o)/p(\mathbf{x}^o)} \\ \vdots \\ \frac{\sum_{\mathbf{x}^o, y \in \mathsf{D}_{\mathsf{train}}} y \cdot p_{\ell_L}(\mathbf{x}^o)/p(\mathbf{x}^o)}{\sum_{\mathbf{x}^o, y \in \mathsf{D}_{\mathsf{train}}} p_{\ell_L}(\mathbf{x}^o)/p(\mathbf{x}^o)} \end{bmatrix}$$

**Regularization.** During parameter learning, it is common to also add a regularization term to the total loss to reduce overfitting. In our case, we use regularizer $L_2(\Theta) = ||\Theta||^2 = \sum_i \theta_i^2$. Now, we want to minimize the following loss:

$$\mathcal{L} = \mathcal{L}_{\mathsf{MSE}} + \lambda L_2 \tag{7}$$

Where $\lambda$ is the regularization hyperparamter. By repeating the steps from above we can easily see that the parameters that minimize $\mathcal{L}$ are:

$$\boldsymbol{\theta}^* = \begin{bmatrix} \frac{\sum_{\mathbf{x}^o, y \in \mathsf{D}_{\mathsf{train}}} y \cdot p_{\ell_1}(\mathbf{x}^o)/p(\mathbf{x}^o)}{\lambda + \sum_{\mathbf{x}^o, y \in \mathsf{D}_{\mathsf{train}}} p_{\ell_1}(\mathbf{x}^o)/p(\mathbf{x}^o)} \\ \vdots \\ \frac{\sum_{\mathbf{x}^o, y \in \mathsf{D}_{\mathsf{train}}} y \cdot p_{\ell_L}(\mathbf{x}^o)/p(\mathbf{x}^o)}{\lambda + \sum_{\mathbf{x}^o, y \in \mathsf{D}_{\mathsf{train}}} p_{\ell_L}(\mathbf{x}^o)/p(\mathbf{x}^o)} \end{bmatrix}$$

$\square$

# B. Expected Parameters Beyond Single Trees

In this section, we extend the expected parameter tuning to beyond single tree models. The learning scenarios include forests, bagging, random forests, and gradient tree boosting.

## B.1. Forests

In this section, instead of a single tree $f_\theta$, we are given a forest $F_\Theta$, and want to minimize the following loss instead:

$$\mathcal{L}_{Forest}(\Theta; D) = \frac{1}{|D|} \sum_{\mathbf{x}^o, y \in D} \mathbb{E}_{p_\Phi(\mathbf{X}^m | \mathbf{x}^o)}\left[l(y, F_\Theta(\mathbf{x}))\right]$$

**Proposition 3** (Expected parameters of forests MSE loss). *Given the training set* $\mathsf{D}_{\mathsf{train}}$, *and given the Forest* $F_\theta$, *the set of parameters* $\Theta$ *that minimizes* $\mathcal{L}_{Forest}$ *can be found by solving for* $\Theta$ *in the following linear system of equations:*

$$M \times \Theta = B \tag{8}$$

*where $M$ is $k \times k$ matrix, $\Theta$ and $B$ are $k \times 1$ vectors.*

$$M[i, j] = \sum_{\mathbf{x}^o, y \in \mathsf{D}_{\mathsf{train}}} p_{\ell_i, \ell_j}(\mathbf{x}^o)/p(\mathbf{x}^o)$$

$$B[i] = \sum_{\mathbf{x}^o, y \in \mathsf{D}_{\mathsf{train}}} y \cdot p_{\ell_i}(\mathbf{x}^o)/p(\mathbf{x}^o)$$

$$\Theta[i] = \theta_i$$

Note that, we usually learn forest, tree by tree and do not have all the tree structures initially, and also above algorithm grows quadratic to number of total leaves which is less desirable. As a result, we also want to explore other scenarios such as bagging or boosting.

## B.2. Bagging and Random Forests

In both Bagging of trees and Random forests, we learn our trees independently and average their predictions, we can also do the expected parameter tuning for each tree independently, w.r.t. a generative model learned on the boostrap sample of the training dataset on which the tree has been induced.

## B.3. Gradient Tree Boosting

In this section, we adapt gradient tree boosting in the expected prediction framework. Before moving on to boosting of trees, we introduce Lemma 2, which computes the expected prediction of two trees multiplied.

**Lemma 2** (Expected tree times tree). *Given two trees* $f_\Theta(\mathbf{x})$, *and* $f'_\Theta(\mathbf{x})$, *a distribution* $p(\mathbf{X})$ *and a partial assignment* $\mathbf{x}^o$ *the expected squared prediction of* $f(\mathbf{x}) \cdot f'(\mathbf{x})$ *w.r.t.* $p$ *can be computed as follows:*

$$\mathbb{E}_{p_\phi(\mathbf{X}^m | \mathbf{x}^o)}\left[f_\theta \cdot f'_\theta\right] = \frac{\sum_{\ell \in \mathsf{leaves}(f)} \sum_{j \in \mathsf{leaves}(f')} \theta_\ell \theta_j p_{\ell, j}(\mathbf{x}^o)}{p(\mathbf{x}^o)}$$

*where* $p_{\ell, j}(\mathbf{x}^o) = p(\mathbf{x}^{\mathsf{path}(\ell)}, \mathbf{x}^{\mathsf{path}(j)}, \mathbf{x}^o)$.

*Proof.* The proof similarly follows from proof of Lemma 1. The main difference is that in $\mathcal{I}_\ell(\mathbf{x}) \cdot \mathcal{I}_{\ell'}(\mathbf{x})$ the leaves $\ell$ and $\ell'$ are from two different trees so its not necessarily equal to 0, so we can not cancel those terms. $\square$

Note that Lemma 2 result can be easily extended to multiplying two forests.

During gradient tree boosting we learn our forest in a additive manner. At each step, given the already learned forest $F$ we add a new tree $f_\theta$ that minimizes sum of losses of the form $l\big(y, F(\mathbf{x}) + f_\theta(\mathbf{x})\big)$. We adapt this with the expected prediction framework as follows:

**Definition 3** (Boosting expected loss minimization). *In addition to definition 2, we are also given a fixed forest $F$, we want to find the set of parameters $\Theta$ of the tree such that $f_\Theta$ minimizes the expected loss $\mathcal{L}_{Boost}$ defined as follows:*

$$\mathcal{L}_{Boost}(\Theta; D) = \frac{1}{|D|} \sum_{\mathbf{x}^o, y \in D} \mathbb{E}_{p_\Phi(\mathbf{X}^m | \mathbf{x}^o)}\Big[l\big(y, F(\mathbf{x}) + f_\Theta(\mathbf{x})\big)\Big]$$

**Proposition 4** (Expected parameters of Boosted MSE loss). *Given the training set $\mathsf{D}_{\text{train}}$, and given the Forest $F$ and the new tree structures $f_\Theta$, the set of parameters $\Theta$ that minimizes $\mathcal{L}_{\text{Boost}}$ can be found by*

$$\theta_\ell^* = \frac{\sum_{\mathbf{x}^o, y \in \mathsf{D}_{\text{train}}} \dfrac{y \cdot p_\ell(\mathbf{x}^o) - \sum_{j \in \text{leaves}(F)} \theta_j p_{\ell,j}(\mathbf{x}^o)}{p(\mathbf{x}^o)}}{\sum_{\mathbf{x}^o, y \in \mathsf{D}_{\text{train}}} \dfrac{p_\ell(\mathbf{x}^o)}{p(\mathbf{x}^o)}}$$

## C. More Experiment Info

*Table 1.* Statistics about the datasets used in the experiments.

| DATASET | TRAIN | VALID | TEST | FEATURES |
|---|---|---|---|---|
| INSURANCE | 936 | 187 | 215 | 36 |

**Description of the datasets** In the INSURANCE [2] dataset, the goal is to predict yearly medical insurance costs of patients given other attributes such as age, gender, and whether they smoke or not.

**Preprocessing Steps** We preserve the original test, and train splits if present for each dataset. Additionally, we merge any given validation set with the test set.

The probabilistic circuit learning implementation that we use does not support continuous features yet, so we perform discretization of the continuous features as follows. First, we try to automatically detect the optimal number of (irregular) bins through adaptive binning by employing a penalized likelihood scheme as in (Rozenholc et al., 2010). If the number of the bins found in this way exceeds ten, instead we employ an equal-width binning scheme capping the bin number to ten. Once the data is discrete, we employ one-hot encoding.

**Other Settings** For XGBoost, we use "reg:squarederror" which corresponds to MSE loss. Max depth is set to 5, and we use regularization $\lambda = 1$ where applicable.

When learning XGBoost trees from missing values, some of the leaves become only reachable if a certain feature is missing, and never reachable with fully observed data. We ignore those leaves in our expected prediction framework.

---

[2]https://www.kaggle.com/mirichoi0218/insurance