# OpenReview forum: "Handling Missing Data in Decision Trees: A Probabilistic Approach"
_ICML.cc/2020/Workshop/Artemiss — ICML Artemiss 2020_

### Official Review · AnonReviewer1 · 2020-06-23
**A probabilistic decision tree to handle missing data**

**Confidence:** 4
**Rating:** 7

**Review:**

The authors proposed to use probabilistic circuits in expected predictions for decision trees. Further, they introduced expected loss minimization that improved imputation while missing values are present at learning time.

The authors mention the weakness of XGBoost, which needs to be trained with missing data to give acceptable performance. However, they used this method in the first experiment, which has a complete training set.

The explanation regarding the median imputation is unclear. Did you consider the median of the training data for the test imputation, or is the median comes from observed test data?

I recommend comparing this method with other baselines such as C4.5,  MICE, and variational autoencoders and add more experiments.

It would be interesting to see computational time and the source code too.

---

### Decision · Program_Chairs · 2020-07-02

**Decision:**

Accept

**Comment:**

We are very happy to inform you that your paper has been accepted for the Artemiss workshop. We will contact you soon to inform you about the details concerning the format of your presentation at the workshop, and the camera-ready version deadline. Please take into account the referee's comments to write the camera-ready version.